# Remote Ischemic Post-Conditioning (RIC) Mediates Anti-Inflammatory Signaling via Myeloid AMPKα1 in Murine Traumatic Optic Neuropathy (TON)

**DOI:** 10.3390/ijms252413626

**Published:** 2024-12-19

**Authors:** Naseem Akhter, Jessica Contreras, Mairaj A. Ansari, Andrew F. Ducruet, Md Nasrul Hoda, Abdullah S. Ahmad, Laxman D. Gangwani, Kanchan Bhatia, Saif Ahmad

**Affiliations:** 1Department of Biology, Arizona State University, Lake Havasu City, AZ 86403, USA; 2Department of Translational Neuroscience, Barrow Neurological Institute, St Joseph’s Hospital and Medical Center (SJHMC), Phoenix, AZ 85013, USAkanchan.bhatia@asu.edu (K.B.); 3Department of Biotechnology, Centre for Virology, Hamdard University, New Delhi 110062, India; 4Department of Neurosurgery, Barrow Neurological Institute, St Joseph’s Hospital and Medical Center (SJHMC), Phoenix, AZ 85013, USA; 5Department of Neurology, Henry Ford Medical Center, Detroit, MI 48202, USA; 6Department of Veterinary Pathobiology, University of Missouri, Columbia, MO 65211, USA; 7School of Mathematical and Natural Sciences, Arizona State University, Glendale, AZ 85306, USA; 8Phoenix Veteran Affairs (VA), Phoenix, AZ 85012, USA

**Keywords:** traumatic optic neuropathy (TON), remote-limb ischemic conditioning (RIC), adenosine monophosphate protein kinase α1 (AMPKα1), inflammation, retinal ganglion cell (RGC), myeloid cell

## Abstract

Traumatic optic neuropathy (TON) has been regarded a vision-threatening condition caused by either ocular or blunt/penetrating head trauma, which is characterized by direct or indirect TON. Injury happens during sports, vehicle accidents and mainly in military war and combat exposure. Earlier, we have demonstrated that remote ischemic post-conditioning (RIC) therapy is protective in TON, and here we report that AMPKα1 activation is crucial. AMPKα1 is the catalytic subunit of the heterotrimeric enzyme AMPK, the master regulator of cellular energetics and metabolism. The α1 isoform predominates in immune cells including macrophages (Mφs). Myeloid-specific AMPKα1 KO mice were generated by crossing AMPKα1^Flox/Flox^ and LysM^cre^ to carry out the study. We induced TON in mice by using a controlled impact system. Mice (mixed sex) were randomized in six experimental groups for Sham (mock); Sham (RIC); AMPKα1^F/F^ (TON); AMPKα1^F/F^ (TON+RIC); AMPKα1^F/F^ LysM^Cre^ (TON); AMPKα1^F/F^ LysM^Cre^ (TON+RIC). RIC therapy was given every day (5–7 days following TON). Data were generated by using Western blotting (pAMPKα1, ICAM1, Brn3 and GAP43), immunofluorescence (pAMPKα1, cd11b, TMEM119 and ICAM1), flow cytometry (CD11b, F4/80, CD68, CD206, IL-10 and LY6G), ELISA (TNF-α and IL-10) and transmission electron microscopy (TEM, for demyelination and axonal degeneration), and retinal oxygenation was measured by a Unisense sensor system. First, we observed retinal morphology with funduscopic images and found TON has vascular inflammation. H&E staining data suggested that TON increased retinal inflammation and RIC attenuates retinal ganglion cell death. Immunofluorescence and Western blot data showed increased microglial activation and decreased retinal ganglion cell (RGCs) marker Brn3 and axonal regeneration marker GAP43 expression in the TON [AMPKα1^F/F^] vs. Sham group, but TON+RIC [AMPKα1^F/F^] attenuated the expression level of these markers. Interestingly, higher microglia activation was observed in the myeloid AMPKα1^F/F^ KO group following TON, and RIC therapy did not attenuate microglial expression. Flow cytometry, ELISA and retinal tissue oxygen data revealed that RIC therapy significantly reduced the pro-inflammatory signaling markers, increased anti-inflammatory macrophage polarization and improved oxygen level in the TON+RIC [AMPKα1^F/F^] group; however, RIC therapy did not reduce inflammatory signaling activation in the myeloid AMPKα1 KO mice. The transmission electron microscopy (TEM) data of the optic nerve showed increased demyelination and axonal degeneration in the TON [AMPKα1^F/F^] group, and RIC improved the myelination process in TON [AMPKα1^F/F^], but RIC had no significant effect in the AMPKα1 KO mice. The myeloid AMPKα1c deletion attenuated RIC induced anti-inflammatory macrophage polarization, and that suggests a molecular link between RIC and immune activation. Overall, these data suggest that RIC therapy provided protection against inflammation and neurodegeneration via myeloid AMPKα1 activation, but the deletion of myeloid AMPKα1 is not protective in TON. Further investigation of RIC and AMPKα1 signaling is warranted in TON.

## 1. Introduction

Traumatic optic neuropathy (TON), a prevalent cause of visual impairment resulting from head or ocular trauma incurred during events such as motor vehicle accidents, natural disasters or wartime activities [1,2], manifests as a spectrum of visual deficits ranging from partial to total blindness. The optic nerve sustains injury directly from penetrating trauma or indirectly from brain-injury-induced phenomena such as hemorrhage, edema or force transmission into the optic nerve [3,4]. Despite the clinical significance, effective treatments for TON-associated optic nerve injury and retinal degeneration remain elusive, with high-dose corticosteroids or decompression surgery showing limited efficacy [5,6]. Remote limb ischemic pre- or post-conditioning (RIC), involving the repetitive inflation–deflation of a blood pressure cuff on one or more limbs, has emerged as a neuroprotective intervention in various brain injury models [7,8,9,10].

Initially employed in a canine myocardial infarction model by Murry et al. (1986) [11], RIC has demonstrated efficacy in clinical trials for myocardial infarction [12], acute ischemic stroke [13], intracranial stenosis and subarachnoid hemorrhage [14]. Additionally, evidence suggests potential protective effects of RIC against retinal ischemia, improving the survival and function of retinal neurons in diabetic retinopathy and glaucoma [15,16]. We reported previously that RIC therapy is protective following TON by preventing retinal ganglion cell (RGC) death, oxidative insult and inflammation in the mouse retina [17], yet the molecular mechanism by which RIC therapy provides neuroprotection remains to be elucidated. Neuroinflammation is a prominent cellular response following injury to the brain, evolving over varying time frames ranging from minutes to years post injury and contributing significantly to both acute and chronic neurological outcomes [18,19,20]. Microglia, the principal immune cells residing in the brain, are believed to play pivotal roles in mediating neuroinflammation [21]. Moreover, peripheral macrophages also actively participate in the acute neuroinflammatory cascade [22]. Early reports indicate that microglia/macrophages exhibit remarkable plasticity, adopting distinct phenotypes in response to different micro environmental cues, including pro-inflammatory and anti-inflammatory signals [23].

The pro-inflammatory M1-like phenotype, characterized by the expression of markers such as CD11 and CD68, tends to release detrimental mediators like tumor necrosis factor-α (TNF-α). In contrast, the anti-inflammatory M2-like phenotype, typified by molecular markers including CD206, generates beneficial mediators such as interleukin-10 (IL-10) and transforming growth factor-β (TGF-β) [24]. Importantly, the M1-like phenotype is often associated with uncontrolled neuroinflammation observed in neurodegenerative diseases, while the M2-like phenotype promotes inflammation resolution and tissue repair [25]. Thus, maintaining a balanced response between polarized microglial/macrophage phenotypes is crucial for immune homeostasis in the brain. In major neurodegenerative diseases, the dualistic roles of distinctly polarized microglial/macrophage populations are reported [26,27]. Animal studies have shown that although both M1-like and M2-like polarized microglial/macrophage populations are activated after brain injury, the M2-like response tends to diminish over time, while the pathological M1-like effect can persist for extended periods post injury [27]. Modulating the balance between M1 and M2 polarization has been indicated to be beneficial for functional outcomes [28,29]. However, the precise molecular mechanisms underlying microglial/macrophage polarization remain incompletely understood. Consistent with early observations, myeloid cells are pivotal in the processes of neurodegeneration observed in multiple neurodegenerative diseases [30,31]. The initial phase of these diseases is marked by the presence of pathogenic activated macrophages, categorized as M1 type, while the subsequent recovery phase is associated with alternatively activated macrophages, referred to as M2 type, which release anti-inflammatory cytokines to resolve the pathogenic inflammation [32,33]. Activated M1 macrophages heavily rely on glycolysis to enhance biosynthetic pathways, facilitating the production of inflammatory mediators [34,35]. Conversely, anti-inflammatory M2 macrophages primarily depend on mitochondrial respiration [36]. 5′ adenosine monophosphate-activated protein kinase (AMPK) serves as a key regulator of cellular energy metabolism, thereby governing the equilibrium between glycolysis and mitochondrial respiration [37,38]. Early studies have demonstrated that AMPKα1 knockout (KO) mice exhibit exacerbated pathology, suggesting a protective role for AMPK activation [39,40,41,42]. AMP-activated protein kinase (AMPK) has shown a protective role in retinal diseases [43,44], and recent evidence suggests that RIC attenuates intracerebral hemorrhage-induced brain injury via AMPK signaling [10]. Given the pivotal role of metabolic alterations in modulating immune effector functions [45], we hypothesized that RIC might ameliorate neurological sequelae through the regulation of immunometabolism subsequent to TON.

## 2. Results

### 2.1. RIC Treatment Preserves Neuronal Cell Death in TON

In vivo funduscopic fluorescein imaging in C57BL/6 mice showed the pathological changes in TON when compared to control subjects characterized by inflammation in the blood vessels suggesting a pronounced vascular response to the traumatic insult. Further investigation through intravenous fluorescein angiography of the mouse retina provided insights into the vascular perfusion dynamics following TON induction. A notable decrease in perfusion, indicative of compromised blood flow through attenuated vasculature, was observed in TON (Figure 1A). The histological examination of the retinal sections stained with H&E provided additional evidence supporting the pathological changes observed in TON. Specifically, a significant increase in neuronal cell death within the ganglion cell layer of mice with TON compared to control animals indicated the location of neuronal loss caused by TON.

After trauma occurs, there is an immediate shearing of a proportion of retinal ganglion cell axons. This initial event is irreversible and leads to neuronal loss. Remarkably, RIC treatment following TON induction resulted in a notable attenuation of neuronal cell death within the ganglion cell layer, suggesting a potential neuroprotective effect of RIC in mitigating the adverse consequences of TON-induced neurodegeneration (Figure 1B).

### 2.2. AMPKα1 Is Essential for RIC-Mediated Reduction of Microglial Activation

TON is characterized by retinal ganglion cell injury, which is implicated in the activation of microglial cells within the mouse retinal model of TON. This activation of microglial cells, in turn, perpetuates a cascade of events, potentially exacerbating retinal damage. Specifically, activated microglial cells may persist in their activation state to engage in phagocytosis of degenerating ganglion cells and release pro-inflammatory cytokines. These cytokines, in particular, contribute to the propagation of inflammatory responses within the retinal microenvironment, ultimately leading to further neuronal death. Therefore, the modulation of microglial activation presents a promising therapeutic intervention in the treatment of TON, with the aim of halting the progression of retinal damage and preserving visual function. We observed a distinct expression of the microglial marker TMEM119 within the mouse retina indicative of microglial activation after TON. Specifically, the induction of TON in animals constitutively expressing AMPK elicited a notable increase in microglial activation. However, RIC treatment led to a significant downregulation of microglial activation in response to TON. Remarkably, a heightened microglial activation was observed in the myeloid-specific AMPKα1 knockout (KO) group compared to the control group after TON. However, the RIC treatment did not exert a significant modulatory effect on microglial activation levels in this group, suggesting the essentiality of AMPK signaling for RIC-mediated neuroprotective mechanisms (Figure 2A,B).

### 2.3. Myeloid AMPKα1 Is Necessary for RIC-Induced Innate Immune Modulation After TON

Microglia, the resident immune cells of the brain, play a pivotal role in the initiation NIH,of the immune response associated with neuropathology. It is widely postulated that these microglial cells are primarily responsible for instigating the immune cascade in response to injury. Elevations in neuroinflammatory markers are consistently observed in individuals with brain injury, as evidenced by numerous studies conducted in both animal models and human subjects. To elucidate the functional role of AMPK in inflammatory responses associated with activated microglia, the expression of CD11b, CD68 and CD206 receptors was compared between AMPK-KO and WT mice 5 days post TON. While the number of pro-inflammatory (CD11b+_CD68+) macrophages in mouse retina increased post injury, no significant differences were observed between Sham and Sham treated with RIC groups on the uninjured side.

On the injured side, RIC treatment significantly reduced injury-dependent pro-inflammatory (CD11b+_CD68+) macrophages, suggesting RIC utility in controlling pro-inflammatory phenotype. However, this beneficial effect of RIC was reversed in myeloid-specific AMPKα1 knockout mice where RIC failed to suppress injury-dependent pro-inflammatory (CD11b+_CD68+) macrophages (Figure 2C–E).

To further investigate the contribution of anti-inflammatory macrophages toward RIC-induced recovery, we analyzed the effect of RIC on the expression of anti-inflammatory (CD11b^+^, F4/80+ and CD206+) macrophages in mouse retina. Whereas TON did not significantly affect the myeloid expression of either CD11b^+^-CD206+ or F4/80+-CD206+ in either WT or myeloid-specific AMPKα1 knockout mice, a RIC-induced increase in the expression of the anti-inflammatory (CD206+ in both CD11b+ and F4/80+) macrophage phenotype was observed following TON in both WT and myeloid-specific AMPKα1 knockout mice. However, these RIC-mediated changes were not observed in Sham-injured mice (not significantly different from mock conditioning), suggesting that the injury-dependent RIC-mediated polarization of anti-inflammatory phenotype is independent of AMPKα1 status (Figure 2F–I).

Next, we analyzed the effect of RIC on the expression of anti-(CD11b+_IL10+, F4/80+-IL10+ and CD68+-IL10+) and pro-inflammatory (CD68+_LY6G+) markers on macrophages and neutrophils in both WT and myeloid-specific AMPKα1-KO mice. Whereas RIC did not significantly affect the myeloid expression of the anti-inflammatory IL10+ marker following TON or TON treated with RIC in myeloid-specific AMPKα-KO mice, a TON-dependent decrease in the expression of IL10+ was significantly reversed with RIC treatment. These RIC-mediated changes were not observed in Sham-injured mice (not significantly different from mock conditioning), nor did RIC induce these effects in myeloid-specific AMPKα1-KO mice (not significantly different from WT injured mice) (Figure 3A–G).

Further analysis of Ly6G, a neutrophil protein that modulates neutrophil migration to sites of inflammation, revealed that RIC did not significantly affect the expression of LY6G, similar to IL-10, in either Sham-operated WT or myeloid-specific AMPKα1 knockout mice after TON. In contrast, RIC induced an AMPKα1-dependent decrease in myeloid LY6G expression following TON (not significantly different from mock conditioning in WT mice) (Figure 3H).

### 2.4. RIC Modulates Pro- and Anti-Inflammatory Cascades in TON via Myeloid AMPKα1 Dependence

We have previously reported an upregulation of the pro-inflammatory cytokine TNF-α and the suppression of anti-inflammatory cytokine IL-10 in retinal tissue after TON [46,47]. The present results elucidate the functional role of AMPK in modulated inflammatory responses associated with activated microglia manifested as the production of pro- and anti-inflammatory cytokines in AMPK-KO and WT mice 5 days post TON. While the levels of pro-inflammatory (TNF-α) cytokine in blood significantly increased post injury, no significant differences were observed between Sham and Sham treated with RIC groups on the uninjured side. In contrast, the levels of anti-inflammatory (IL-10) cytokine in blood decreased significantly post injury, and no significant differences were observed between Sham and Sham treated with RIC groups on the uninjured side. On the injured side, RIC treatment significantly reduced injury-dependent pro-inflammatory (TNF-α) cytokine production, suggesting RIC utility in controlling pro-inflammatory phenotype. However, this beneficial effect of RIC was reversed in myeloid-specific AMPKα1 knockout mice where RIC failed to suppress injury-dependent pro-inflammatory (TNF-α) cytokine (Figure 4A). Notably, RIC treatment significantly increased anti-inflammatory (IL-10) cytokine production (not significantly different from the mock-treated Sham). However, this beneficial effect of RIC was reversed in myeloid-specific AMPKα1 knockout mice where RIC failed to increase injury-dependent anti-inflammatory (IL-10) cytokine production (Figure 4B). Our results indicate that the modulated expression of pro- and anti-inflammatory cytokines involves the activation of AMPKα1 in myeloid cells. Upon activation, AMPKα1, a key regulator of cellular energy homeostasis, suppresses the activity of pro-inflammatory pathways while promoting the expression of anti-inflammatory mediators, tilting the balance towards a less inflammatory environment. RIC treatment likely activates AMPKα1 in myeloid cells, leading to the downregulation of pro-inflammatory cytokines such as TNF-α and the upregulation of anti-inflammatory cytokines like IL-10, attenuating the inflammation and contributing to the therapeutic effects of RIC in TON.

Earlier reports suggest that ischemic injury leads to retinal ganglion cell (RGC) death leading to the loss of vision [48,49]. We next assessed the impact of RIC on preserving the RGCs by controlling the inflammatory processes through regulating ICAM-1 expression. The results further cement the critical role of AMPK in immunomodulation. The immunofluorescence analysis of ICAM-1 expression in retinal section revealed significantly increased (>2 fold) ICAM-1 expression post injury, and no significant differences were observed between Sham and Sham treated with RIC groups on the uninjured side. Consistent with pro-inflammatory cytokine results in blood, the ICAM-1 expression was attenuated significantly with RIC treatment post injury (not significantly different from the Sham and mock-treated Sham). However, this beneficial effect of RIC was reversed in myeloid-specific AMPKα1 knockout mice where RIC failed to suppress injury-dependent ICAM-1 expression (Figure 5A,B).

Furthermore, Western blot analysis revealed a similarly upregulated expression of ICAM-1 in retinal tissue from AMPK-KO and WT mice 5 days post TON. RIC treatment following TON significantly decreased (~33%) ICAM-1 expression. However, the beneficial effects of RIC were completely reversed in myeloid-specific AMPKα1 knockout mice (Figure 5C,D). Taken together, these results suggest that AMPKα1 is crucial for the RIC-mediated attenuation of retinal inflammation and neuronal cell injury associated with TON.

#### RIC Treatment Induced Retinal Oxygenation Is AMPK Independent in TON

The propagation of shock waves resulting from concussive forces directed at the head transmits through the optic canal. Early reports indicate that optic nerve swelling within the confined space of the optic canal compromises blood supply and exacerbates tissue ischemia, further damaging the injured optic nerve [4,50]. Vascular insufficiency likely contributes significantly to TON pathophysiology. Thus, we tested the hypothesis that RIC therapy decompresses the optic canal by increasing blood supply to the optic nerve and retina. Our results show that RIC treatment increased retinal oxygenation in Sham animals, which was significantly different from the mock-treated Sham. Among the injured groups, retinal oxygen levels were depleted by 55% in TON eyes compared to controls, both in WT and myeloid-specific AMPKα1 knockout mice, potentially attributed to decreased blood flow in the eye. Notably, RIC treatment improved oxygen levels by 44% in TON eyes in both the groups, namely, WT and myeloid-specific AMPKα1 knockout mice (Figure 6A,B). The restored retinal tissue oxygenation among AMPK-WT was not significantly different when compared with that of myeloid-specific AMPKα1 knockout mice, suggesting that RIC-treatment-induced retinal oxygenation is independent of the AMPK genotype (Figure 6A–D). In order to elucidate the molecular mechanism behind this phenomenon, we analyzed the mitric oxide (NO), an intracellular signaling molecule well known for its key role in vasodilation, levels in the plasma of all experimental groups. Early reports have shown diverse ocular effects of NO in both preclinical models and clinical studies [51,52,53]. Consistent with retinal tissue oxygen tension, NO levels were found depleted (~50%) significantly after TON when compared with the control, which were restored significantly (~40%) after RIC treatment (Figure 6C). Our results provide empirical evidence of RIC being a potential non-invasive therapy for retinal vascular pathophysiological changes in TON.

### 2.5. RIC Preserves Retinal Ganglion Cells and Rescues Myelin Integrity via AMPK Activity in TON

We have previously shown a high level of cell death localized within the ganglion cell layer, 5 days post TON, in both the murine optic nerve crush model [47] and crush optic injury (COI) [17], which was significantly lowered after RIC treatment. In order to elucidate the mechanism, we analyzed the effect of AMPK1a genotype on Brn3 and growth-associated protein 43 (GAP-43) expression in WT and myeloid-specific AMPKα1 knockout mice with or without RIC treatment after TON. The Brn3 family of POU-domain transcription factors play important roles in differentiation, survival and axonal elongation during the development of murine RGCs [54]. GAP43, a pivotal constituent of the neural growth cone in mammals, contributes significantly to neurodevelopment and synaptic function [55] and also mediates nerve regeneration subsequent to nerve injury. Whereas RIC did not significantly affect the retinal expression of either the retinal ganglion cell (RGC) marker Brn3 or the axonal growth-associated protein GAP43 following TON in either myeloid-specific AMPKα1 knockout mice, with or without RIC treatment (not significantly different from AMPK-WT mice), a TON-independent increase in the expression of both Brn3 (*p* < 0.01 vs. AMPK-WT mice as well as myeloid-specific AMPKα1-KO and *p* < 0.05 vs. myeloid-specific AMPKα1-KO+ RIC mice) (Figure 7A,B) and GAP43 (*p* < 0.01 vs. AMPK-WT mice as well as myeloid-specific AMPKα1-KO and myeloid-specific AMPKα1-KO+ RIC mice) (Figure 7C,D) was observed in AMPK-WT mice following RIC.

As a result of optic nerve injury, a multifaceted metabolic, cellular and structural response renders intact neurons and glia near the injury site susceptible to secondary degeneration, exacerbating functional loss. Therefore, we analyzed the effect of RIC on axonal and myelin integrity through ultrastructural analyses of surviving optic nerve tissue undergoing secondary degeneration five days post TON. No ultrastructural changes were observed in Sham-injured mice (not significantly different from mock conditioning), nor did RIC induce any effects in Sham-injured animals. However, significant morphological loss in RGC axons and associated demyelination was observed in AMPK-WT mice after TON when compared with Sham group. Notably, RIC treatment preserved RGC axons, myelin sheaths, fiber diameter and myelin thickness significantly in AMPK-WT mice. In contrast, these RIC-mediated ultrastructural changes were not observed in myeloid-specific AMPKα1 knockout mice (not significantly different from myeloid-specific AMPKα1 knockout mice without RIC treatment) (Figure 8). Thus, our data demonstrate that myeloid-specific AMPKα1 knockout abolishes the benefits of RIC therapy, at least in the early phase of TON, with or without RIC.

## 3. Discussion

Our study, for the first time, shows the role of AMPKα1 in neuroprotective effects of RIC following TON in preserving neuronal integrity and modulating inflammatory responses in mouse eyes. Traumatic injury to the optic nerve triggers a series of interconnected events including impaired axonal transport [56], localized inflammatory responses [46,57], excitotoxicity [58], oxidative stress [59] and DNA damage [60], collectively driving Wallerian degeneration, RGC dysfunction [61] and, ultimately, cell death [62,63,64]. Due to their inability to regenerate, apoptotic retinal ganglion cells (RGCs) lead to permanent vision loss.

### 3.1. Remote Ischemic Conditioning (RIC) Has Neuroprotective Role in TON

We have previously shown that RIC therapy protects against TON via AMPKα1 activation [17], a catalytic subunit of the enzyme AMPK, which regulates cellular energy and metabolism. The α1 isoform, predominant in immune cells such as macrophages (Mφs), is crucial for RIC-induced anti-inflammatory macrophage polarization, suggesting a molecular link between RIC and immune activation [10,65,66]. The injury to white matter tracts, such as the optic nerve, triggers secondary degeneration processes that lead to further neuronal damage, extending beyond the initial site of injury [67,68,69]. In the case of TON, pathological changes, including inflammation and compromised blood flow within retinal blood vessels, impair vascular perfusion, exacerbating secondary degeneration, as decreased oxygen and nutrient supply can promote neuronal cell death. Our histological analyses confirmed significant neuronal loss in the ganglion cell layer, a hallmark of secondary degeneration that was attenuated after RIC treatment, in turn protecting retinal ganglion cells, reducing neuronal loss and mitigating the harmful effects of secondary degeneration in TON. The protective effects of RIC in TON against secondary degeneration likely arise from its ability to modulate key processes involved in inflammation, blood flow and neuronal survival. The injury to white matter tracts, such as the optic nerve, triggers a cascade of inflammatory events and disrupts blood flow, leading to compromised oxygen and nutrient delivery to RGCs. This ischemic environment exacerbates secondary degeneration by promoting oxidative stress and metabolic dysfunction, which are known to accelerate neuronal death.

RIC likely mitigates these processes by improving vascular perfusion and reducing inflammation. Studies have shown that RIC enhances NO production, which promotes vasodilation and improves blood flow to the damaged areas. In the context of TON, improved blood flow can counteract the ischemic conditions that contribute to neuronal damage. Moreover, RIC has been demonstrated to downregulate pro-inflammatory cytokines, such as TNF-α, and upregulate anti-inflammatory mediators like IL-10, which can reduce the inflammatory burden in the optic nerve and retinal tissues. This reduction in inflammation may protect RGCs from further damage caused by infiltrating immune cells and the inflammatory cytokines they release.

Histological evidence showing that RIC treatment attenuates neuronal loss in the ganglion cell layer suggests that RIC interrupts the vicious cycle of inflammation and ischemia that drives secondary degeneration. By protecting RGCs and enhancing retinal blood flow, RIC not only reduces the extent of neuronal death but also creates an environment conducive to neuronal survival. This indicates that RIC’s effects are multifaceted—targeting both the vascular and inflammatory components of secondary degeneration—making it an effective strategy for mitigating neuronal loss in TON. The absence of these protective effects in untreated or control mice reinforces the argument that RIC directly influences the key mechanisms underlying secondary degeneration in TON. Neurological pathologies are commonly driven and exacerbated by an inflammatory response. Microglial activation occurring early after injury [70,71] often coincides with neuronal degeneration and axonal abnormalities [72], suggesting that inflammation may contribute to the development of neurodegenerative disorders.

Thus, our study further elucidated the mechanisms of RIC-mediated neuroprotection focused on the activation of microglial cells. It was observed that TON induced a substantial increase in microglial activation, as indicated by the expression of the microglial marker TMEM119. Notably, RIC significantly downregulated microglial activation in wild-type (WT) mice, but this effect was absent in myeloid-specific AMPKα1 knockout (KO) mice, indicating that AMPK signaling is crucial for the neuroprotective effects of RIC. Here, it is important to mention that retinal flatmount analysis could give us clearer results in terms of microglial activation and RGC death so there is limitation in data presentation. 

### 3.2. Myeloid AMPKα1 Regulates Anti-Inflammatory Signaling Following TON

Early studies have shown that AMPKα1 is vital for IL-10 activation during macrophage polarization, which is mediated through the PI3K/AKT/mTORC1 and STAT3 pathways [73]. Its rapid activation implies that AMPK serves as an upstream signaling molecule in the initiation of anti-inflammatory pathways [73]. We explored the role of myeloid AMPKα1 in RIC-mediated immune modulation by analyzing the expression of pro- and anti-inflammatory markers in macrophages. RIC significantly reduced the number of pro-inflammatory (CD11b^+^-CD68^+^) macrophages post TON in WT mice, while this effect was abolished in AMPKα1 KO mice, underscoring the necessity of AMPK signaling in RIC-mediated immune modulation. The absence of AMPKα1 enhances the proinflammatory activity of myeloid antigen-presenting cells (APCs) and intensifies CD40 signaling [74]. Interestingly, RIC increased the expression of anti-inflammatory (CD206^+^) macrophages, independent of AMPKα1 status, suggesting that some RIC effects may bypass AMPK signaling. The observed increase in anti-inflammatory (CD206^+^) macrophages following RIC, independent of AMPKα1 status, suggests that RIC may engage alternative signaling pathways to modulate immune responses. While AMPKα1 is critical for regulating macrophage polarization, particularly in promoting anti-inflammatory responses via the PI3K/AKT/mTORC1 and STAT3 pathways, the fact that RIC can still induce CD206+ macrophages in AMPKα1 KO mice indicates the involvement of additional or compensatory mechanisms. One plausible explanation is that RIC could activate other upstream regulators of macrophage polarization, such as the IL-4/STAT6 pathway, which is known to induce CD206^+^ expression and anti-inflammatory macrophage phenotypes. Additionally, RIC might enhance the activity of other kinases or transcription factors, such as PPARγ, which are also implicated in driving anti-inflammatory responses independent of AMPK. The involvement of nitric oxide (NO), which was shown to increase post RIC, could also be significant. NO has been implicated in promoting anti-inflammatory macrophage profiles and could play a role in the observed upregulation of CD206^+^ macrophages without the need for AMPKα1. However, the failure of RIC to reduce pro-inflammatory (CD11b^+^-CD68^+^) macrophages in AMPKα1 KO mice underscores the necessity of AMPK signaling in controlling the pro-inflammatory arm of the immune response. Without AMPKα1, myeloid APCs may retain their heightened inflammatory state, likely driven by unmitigated CD40 signaling, which intensifies the pro-inflammatory activity of these cells. This dichotomy—RIC’s ability to upregulate anti-inflammatory macrophages but not downregulate pro-inflammatory ones in the absence of AMPKα1—points to a complex interplay of signaling pathways. It suggests that while RIC can partially bypass AMPKα1 to promote anti-inflammatory responses, it relies heavily on AMPKα1 to suppress pro-inflammatory activity and achieve full immune modulation. Thus, AMPKα1 remains a key determinant of RIC’s overall immunomodulatory efficacy.

Early studies have demonstrated elevated levels of TNF-α and Iba-1 after TON [47], and TNF-α inhibition has been shown to provide protection during TON [75]. Additionally, microglia have been identified as the primary source of proinflammatory cytokines, such as TNF-α, which mediates RGC death through its receptor TNF-R1 in both TON and retinal ischemia [75,76,77]. Moreover, IL-10 has been found to play a protective role by reducing inflammation in various CNS diseases, including retinal conditions [78,79]. Our RIC treatment reduced the expression of the pro-inflammatory cytokine TNF-α and increased the anti-inflammatory cytokine IL-10 in WT mice post TON. These beneficial effects were not observed in AMPKα1 KO mice, further emphasizing the critical role of AMPKα1 in RIC-induced immunomodulation. The absence of beneficial effects in AMPKα1 KO mice post TON, particularly the lack of TNF-α reduction and IL-10 increase, suggests that AMPKα1 plays a pivotal role in RIC-induced immunomodulation. AMPK is known to regulate metabolic and inflammatory pathways, and its activation has been shown to suppress proinflammatory signals such as TNF-α while promoting anti-inflammatory mediators like IL-10. Without AMPKα1, RIC may fail to activate these regulatory pathways, allowing TNF-α levels to remain elevated and inhibiting the upregulation of IL-10. The argument for AMPKα1’s involvement becomes stronger when considering its role in modulating microglial activity. Microglia, the primary source of TNF-α in the retina post injury, are likely influenced by AMPK signaling. In AMPKα1 KO mice, microglia could remain in a heightened proinflammatory state, exacerbating RGC death through TNF-R1-mediated pathways. In contrast, AMPK activation in WT mice may shift microglial activity towards a neuroprotective profile, characterized by the release of IL-10, which has well-established protective roles in CNS inflammation. Therefore, the failure of RIC to elicit these cytokine changes in AMPKα1 KO mice underscores the enzyme’s essential function in controlling the inflammatory environment, particularly in balancing the harmful effects of TNF-α and the protective effects of IL-10. This suggests that AMPKα1 is indispensable for RIC’s immunomodulatory effects as it governs the molecular mechanisms necessary for reducing inflammation and promoting neuronal survival. In retinal tissues, ICAM-1 acts as a key adhesion molecule, and its increased expression is often considered a marker for inflammation and vascular injury [80], particularly in conditions where leukocyte infiltration into the retina is prominent; essentially, high ICAM-1 levels indicate the potential for immune cells to readily adhere to the retinal blood vessels, contributing to the inflammatory process [81]. A recent study has shown that the inhibition of ICAM-1 gene transcription can reduce its expression in inflamed retinal tissues, thereby mitigating inflammation [81].

Chronic inflammation in retinal tissues, driven by ICAM-1-mediated leukocyte adhesion and transmigration, can exacerbate neuronal damage, particularly in RGCs. The link between inflammation and neurodegeneration is well documented, with inflammatory cytokines, such as TNF-α, promoting RGC death in conditions like glaucoma and retinal ischemia [82]. When an injury occurs to the optic nerve like in traumatic optic neuropathy, activated glial cells in the affected area release pro-inflammatory cytokines like TNF-α, which directly signal RGCs to undergo apoptosis [76]. Elevated ICAM-1 expression is, therefore, not only a marker for vascular injury but also a critical factor in the neuroinflammatory pathways that contribute to the progressive degeneration of neural tissues in the retina. Studies have shown that inhibiting ICAM-1 can reduce leukocyte adhesion, diminish inflammation and protect against neurodegeneration [83,84], highlighting its role in both vascular and neuronal integrity. Presently, RIC treatment significantly reduced the expression of the adhesion molecule ICAM-1 in retinal tissues; however, this effect was reversed in AMPKα1 KO mice, highlighting the dependence of RIC’s protective effects on AMPKα1. The reversal of RIC’s protective effects on ICAM-1 expression in AMPKα1 KO mice suggests that AMPKα1 plays a key role in mediating the anti-inflammatory and neuroprotective effects of RIC. AMPK is a critical energy-sensing enzyme that regulates cellular responses to stress, including inflammation. In retinal tissues, AMPKα1 activation likely dampens the inflammatory response by inhibiting pathways that lead to the upregulation of adhesion molecules like ICAM-1. Without AMPKα1, these anti-inflammatory effects may be diminished, resulting in the persistence of ICAM-1 expression and continued leukocyte adhesion and transmigration, which exacerbate inflammation and neuronal damage. Furthermore, AMPKα1 might regulate ICAM-1 expression through its influence on downstream signaling pathways, such as NF-κB or MAPKs, which are known to control the transcription of pro-inflammatory genes, including ICAM-1. In AMPKα1 KO mice, the absence of this regulatory mechanism could leave these pathways unchecked, leading to higher levels of ICAM-1 and inflammation. The dependence of RIC’s protective effects on AMPKα1 could also involve its role in modulating oxidative stress, another factor linked to both ICAM-1 expression and neurodegeneration. Therefore, the loss of AMPKα1 disrupts the balance of pro- and anti-inflammatory signaling in the retina, weakening the beneficial effects of RIC.

### 3.3. Remote Ischemic Conditioning (RIC) Improves Retinal Oxygenation via AMPKα1

Oxygen (O_2_) is vital for retinal function, diffusing passively from the circulation and used for ATP production. Unique aspects of retinal oxygenation, such as dual circulation, the lack of metabolic regulation in the choroid, O_2_ regulation in retinal circulation, and the concentration of mitochondria in photoreceptor inner segments, influence retinal disease pathogenesis and treatment. Additionally, the retina is one of the most metabolically active tissues, consuming O_2_ faster than many others [85], including the brain [86]. Few studies have shown the efficacy of improved retinal oxygenation among human patients with direct [87] and indirect [88] TON, suggesting that improved oxygenation may constitute a promising therapeutic option for vision recovery. We also explored the impact of RIC on retinal oxygenation, revealing that RIC improved retinal oxygen levels in both WT and AMPKα1 KO mice post TON, indicating that this aspect of RIC’s protective effects may be independent of AMPK signaling. This improvement in oxygenation was accompanied by a significant increase in nitric oxide (NO) levels, which are known to play a role in vasodilation and blood flow regulation [89]. The improvement in retinal oxygenation observed with RIC treatment, even in AMPKα1 KO mice, suggests that mechanisms other than AMPK signaling are involved in enhancing retinal blood flow and oxygen delivery. One plausible explanation is the increased production of nitric oxide (NO) following RIC, which promotes vasodilation. NO is a well-known regulator of blood vessel relaxation and could enhance retinal perfusion by widening blood vessels and improving oxygen supply to the injured tissue. This vasodilatory effect could compensate for the absence of AMPK signaling in KO mice, contributing to the observed rise in retinal oxygen levels and supporting RIC’s neuroprotective role in TON, particularly through improved vascular function. Other potential pathways could involve alternative stress-response mechanisms that are triggered by ischemic conditioning, such as the activation of hypoxia-inducible factors (HIFs) or the modulation of oxidative stress pathways, which might independently enhance tissue oxygenation and protect against retinal damage.

### 3.4. Remote Ischemic Conditioning (RIC) Attenuates Retinal Neuronal/Axonal Death via AMPKα1 Following TON

RGC phenotypic markers like Brn3a are commonly used to identify surviving RGCs [90], while regeneration-associated proteins like GAP-43 assess RGC axon regeneration [91]. RIC treatment, in our study, led to a significant increase in Brn3 and GAP-43 expression in WT mice following TON, demonstrating improved RGC survival and axonal regeneration. In contrast, the absence of these effects in AMPKα1 KO mice strongly argues that AMPK signaling is essential for RIC-mediated neuroprotection, emphasizing the critical role of this pathway in the therapeutic response. Without AMPK activation, the benefits of RIC are diminished, highlighting its importance in neurodegenerative processes. The absence of enhanced RGC survival and axonal regeneration in AMPKα1 KO mice likely suggests that AMPK signaling may be a key regulator of cellular energy homeostasis, stress response and survival mechanisms in RGCs. AMPK could influence neuroprotection by promoting mitochondrial function, reducing oxidative stress and modulating the inflammatory pathways that are critical in post-injury recovery. RIC likely activates AMPK, triggering downstream pathways like autophagy, axonal repair and neurotrophic support, which are essential for the observed neurodegeneration in WT mice. Without AMPK, these protective mechanisms may be impaired, leading to the reduced efficacy of RIC. The absence of RIC-mediated ultrastructural protection in myeloid-specific AMPKα1 knockout mice provides compelling evidence that AMPKα1 within myeloid cells is critical for the neuroprotective effects of RIC in optic nerve injury. The preservation of axonal and myelin integrity in AMPK-WT mice post TON suggests that AMPKα1 plays a pivotal role in mediating cellular responses that prevent secondary degeneration. In contrast, the lack of protective effects in myeloid-specific AMPKα1 knockout mice points to myeloid cells—particularly microglia and macrophages—as central players in this process. It is well documented that after optic nerve injury, activated microglia and infiltrating macrophages contribute to both neuroinflammation and repair processes. AMPKα1 likely acts as a molecular switch in these cells, promoting anti-inflammatory and neuroprotective pathways that preserve neuronal and myelin integrity. Without AMPKα1, myeloid cells may fail to shift from a pro-inflammatory to a reparative phenotype, leading to the unchecked secondary degeneration seen in knockout mice. Moreover, RIC’s inability to protect axons and myelin in these knockout mice strengthens the argument that AMPKα1 signaling in myeloid cells is essential for the therapy’s efficacy. RIC likely triggers pathways dependent on AMPKα1 that modulate cellular metabolism, reduce oxidative stress and promote repair mechanisms. The lack of morphological preservation in knockout mice underscores the hypothesis that myeloid AMPKα1 is required for initiating and sustaining the cellular processes that mitigate the damage and foster recovery after optic nerve injury. Thus, the failure of RIC to preserve axonal and myelin integrity in the absence of AMPKα1 emphasizes the enzyme’s indispensable role in mediating the neuroprotective response, suggesting that myeloid AMPKα1 is a key regulator of the reparative mechanisms that RIC activates post TON.

The precise mechanism by which transient ischemia, such as that induced by RIC, alters immunometabolism, in TON, remains elusive. However, shear stress resulting from blood pressure cuff inflation is likely a key mediator of the observed effects. This raises critical questions: Are the benefits of RIC, such as enhanced AMPK activation, directly impacting immune cells, or are intermediary cell types, possibly endothelial cells, playing a role? The association of plasma nitrite with increased AMPK activation provides a compelling argument for the involvement of the vasculature in mediating these immunoregulatory effects. Nitrite, a known signaling molecule in vascular biology [92,93], may enhance AMPK activity indirectly by improving endothelial function and blood flow, which in turn could influence immune cell behavior in the context of injury [94]. This mechanism seems particularly relevant in TON, where vascular damage and impaired retinal oxygenation exacerbate neuronal death. By improving circulation through vasodilation, nitrite and AMPK activation may help to attenuate inflammation and promote neuroprotection. However, in the specific case of TON, where the optic nerve is injured and microglial activation is heightened, the vasculature’s role might extend beyond mere blood flow regulation. Endothelial cells lining the retinal vasculature could be crucial intermediaries, modulating the inflammatory response via AMPK signaling pathways that control cytokine release. This mechanism may also help to explain the varied responses to RIC observed in WT and AMPKα1 KO mice, where the latter lacks the necessary AMPK activation to trigger these protective pathways. To refine RIC as a treatment for TON, further investigation into the exact cellular mechanisms—whether directly affecting immune cells or mediated by endothelial cells—will be necessary. This knowledge could inform the optimization of RIC parameters, such as frequency, duration, and cycle number, to maximize its neuroprotective potential in conditions like TON, where the precise modulation of immune and vascular responses is critical for vision recovery.

## 4. Materials and Methods

### 4.1. Experimental Groups and Ethical Procedures

C57BL/6J wild-type 10-week-old mice were obtained from Jackson Laboratory (Bar Harbor, ME, USA Stock# 000664) or from our in-house colony. The approval of the Institutional Animal Care and Use Committee (IACUC protocol No. 605) of St. Joseph’s Hospital and Medical Center (SJHMC) was obtained prior to the animal experiments and was in accordance with the ARVO (Association for Research in Vision and Ophthalmology) statement for the use of animals in ophthalmic and vision research. The animals were group-housed in a temperature-controlled environment (12 h light/dark cycles; 23 °C; 40–50% humidity) with no more than five mice in a cage bedded with aspen wood chip material and provided with free access to water and chow. The animals (25 g body weight) were acclimated for 7 days prior to the start of the experiment. Animals were sacrificed by cervical dislocation after deep anaesthetization with isoflurane (2–3%). Mice (mixed sex) were randomized in 6 experimental groups for Sham (mock, AMPKα1^F/F^); Sham (RIC, AMPKα1^F/F^); AMPKα1^F/F^ (TON); AMPKα1^F/F^ (TON+RIC); AMPKα1^F/F^ LysM^Cre^ (TON); AMPKα1^F/F^ LysM^Cre^ (TON+RIC). RIC therapy was given every day (5–7 days following TON).

### 4.2. Generation of Mice with Myeloid-Specific AMPKα1 Knockout

Myeloid-specific AMPKα1 KO mice were generated by crossing AMPKα1^Flox/Flox^ and LysM^cre^ to carry out the study. Briefly, LysM^Cre^ mice (B6.129P2-Lyz2tm1(cre)lgo/J; Jackson Laboratories, stock No. 004781) carrying a nuclear localized Cre recombinase inserted into the first coding ATP of the lysozyme 2 gene were crossed with AMPKα1f/f mice (Prkaa1tm1/1Sjm/J: Jackson Laboratories, stock No. 014141), possessing loxP sites flanking exon 3 of the AMPKα1 gene. The AMPKα1f/f mice were subjected to at least 10 generations of backcrossing to C57BL/6J mice. Successful mating was confirmed via PCR genotyping and flow cytometry analysis of AMPK expression in myeloid cells. Littermates were utilized in all experimental procedures.

### 4.3. Traumatic Optic Neuropathy (TON)

We induced TON in mice by using a controlled impact system. A commercially available controlled impact system device (PinPoint PCI3000 Precision Cortical Impactor; Hatters Instruments, Cary, NC, USA) integrated with PCI3000 software was utilized to establish the TON model, following an earlier reported procedure by our group [95]. Briefly, mice were anesthetized using isoflurane (3% for induction and 1.5% during surgery, mixed with 30% O_2_/70% N_2_O) and positioned in a stereotaxic apparatus (Stoeltin Co, Wood Dale, IL, USA) while maintaining body temperature at 37  ±  0.5 °C. An incision was made at the medial canthus of each mouse using laminectomy forceps and scissors. The globe was then gently retracted from the orbital margin with a noninvasive retractor, exposing the extraocular tissue for controlled impact by a blunt impactor tip with a 1 mm diameter. The impactor was advanced to the zero point, ensuring the tip touched the surface of the impact site, before triggering the impact switch to induce trauma. Controlled optic trauma was inflicted at velocities of either 2.0 or 3.0 m/s, with the contusion depth and time held constant at 6.0 mm and 100 ms, respectively. The right eye was selected for optic nerve trauma induction. A naïve control group underwent anesthesia and surgery but was not subjected to impact. The globe or retina was harvested for biochemical analysis five days post TON.

### 4.4. Remote Ischemic Conditioning (RIC)

An automated blood pressure instrument equipped with a blood pressure cuff, specifically the Hatters Instrument from NC, USA, was employed for RIC. RIC procedures were conducted following established protocols utilized in prior investigations involving murine models of stroke, Vascular Cognitive Impairment and Dementia (VCID) and intracerebral hemorrhage (ICH) [8,9,10]. Briefly, customized mouse limb cuffs were applied to each hind limb, and RIC was administered with a regimen consisting of 3 cycles lasting 5 min per cycle, each at a pressure of 150 mmHg, with a 5 min reperfusion interval. Throughout the procedure, the core temperature of the mice was maintained at 37 °C. The rationale behind remote limb conditioning is to activate tissues that exhibit greater resilience to sub-lethal ischemic events, such as limb muscles, as well as tissues or organs that are remote from the actual site of injury. In the context of our study involving ocular trauma, the hind limb represents the most distal site with substantial muscle mass. Hence, RIC was applied to the hind limbs, which are commonly targeted sites for RIC in both animal models and humans alike.

### 4.5. Fluorescein Angiography (FA)

FA was performed according to the previously published method [96]. Briefly, we used 2% isoflurane to anesthetize the mice, and, to dilate the eye pupils, 1% tropicamide eye drops were used. Experimental mice were placed on the imaging platform of the Phoenix Micron III retinal imaging microscope, supplemented with an OCT imaging device (Phoenix Research Laboratories, Pleasanton, CA, USA). For mice eye imaging, 10% fluorescein sodium (Apollo Ophthalmics, Newport Beach, CA, USA) was injected with (10–20 µL, IV), followed by the rapid acquisition of fluorescent images for ~5 min. Fluorescein leakage showed hazy fluorescence and inflamed vasculature.

### 4.6. Immunofluorescence

Immunofluorescence analysis was performed using frozen retinal sections. We utilized our previous published protocol [47]. Briefly, these sections, measuring 10 μm in thickness, were initially fixed in 4% paraformaldehyde, followed by blocking with 10% normal goat serum (NGS). Subsequently, sections were subjected to overnight incubation at 4 °C with primary antibodies, namely, mouse anti-TREM119 (obtained from Santa Cruz Biotechnology, Dallas, TX, USA). On the subsequent day, sections underwent a triple wash with 0.1% PBS-T for 10 min each, followed by a 1 h incubation at room temperature with FITC-conjugated secondary antibody (procured from Invitrogen, Carlsbad, CA, USA). Post incubation, sections were briefly rinsed with PBS-T and then mounted with DAPI (Vector Lab, Newark, CA, USA). For analysis, sections (5 fields/retina, *n* = 5 in each group) were observed under fluorescence microscopy (Keyence, Japan), with Image J software (NIH, https://imagej.net/ij/) employed for quantification of immunostaining intensity

### 4.7. Hematoxylin and Eosin (H&E) Staining

Longitudinal sections of the optic nerve (10 µm thick, three sections per animal, *n* = 5/group) were stained using H&E (Merck, Darmstadt, Germany). Imaging was performed with an Axio Imager M1 microscope (Zeiss, Oberkochen, Germany) at a 400× magnification, capturing three images of each cryosection of mouse retina.

### 4.8. Western Blotting

We performed Western blotting experiments according to our earlier published method [46]. Briefly, retinal tissue was used to prepare whole cell lysates using modified RIPA buffer (Upstate, Lake Placid, NY, USA), comprising 50 mmol/L Tris, 150 mmol/L NaCl, 1 mmol/L ethylenediaminetetra acetic acid, 1% nonidet P-40 and 0.25% deoxycholate, supplemented with 40 mmol/L NaF, 2 mmol/L Na_3_VO_4_, 0.5 mmol/L phenylmethylsulfonyl fluoride and 1:100 (*v*/*v*) of proteinase inhibitor cocktail (Sigma-Aldrich, MO, USA). Centrifugation at 12,000× *g* at 4 °C for 30 min was used to remove insoluble material before quantification of the protein using the DC Protein Assay (Bio-Rad, Hercules, CA, USA) and subsequent boiling of 50–100 μg of protein in Laemmli sample buffer. Separation was achieved via SDS-PAGE on a gradient gel (4 to 20%) (Pierce, Rockford, IL, USA), followed by transfer to PVDF membrane and incubation with specific antibodies. Antibodies targeting β-actin (Sigma), Brn3a (Santacruz Biotechnology, Paso Robles, CA, USA) and GAP43 were utilized, with detection facilitated by the respective horseradish peroxidase-conjugated secondary antibodies and visualization through ECL chemiluminescence (Amersham BioSciences, Buckinghamshire, UK). Densitometry analysis was performed using Image J software (NIH, https://imagej.net/ij/).

### 4.9. Analytical Flow Cytometry

Blood samples (200 µL) were obtained either via cardiac puncture or through the retro-orbital sinus. The collected cells were incubated with antibodies targeting various conjugated cell surface markers: CD11b (BD Biosciences, 557396, Clone M1/70), F4/80 (BD Biosciences, 565613, Clone T45-2342), CD206 (BD Biosciences, Cat 565250, Clone MR5D3), CD68 (BioLegend, 137010, Clone FA-11) and Ly-6G (BioLegend, 127603, Clone 1A8). Subsequently, cells were fixed and permeabilized using a Fixation/Permeabilization Concentrate (Affymetrix eBioscience, Santa Clara, CA, USA) and then incubated with antibodies for intracellular labeling of IL-10 (BioLegend, 505009, Clone JES5-16E3). After a final wash, flow cytometric analysis was conducted using a four-color flow cytometer (FACSCalibu, BD Biosciences) and CellQuest software (BD Biosciences, Franklin Lakes, NJ, USA), as previously described [97]. Isotype-matched controls were employed to establish appropriate gating for each sample. Duplicate analyses were performed for each marker. To reduce false-positive events, the number of double-positive events detected with isotype controls was subtracted from the number of double-positive cells stained with corresponding antibodies. Viable cells were distinguished from debris by gating on live cells exhibiting high forward scatter (FSC) and positivity for specific antibodies. Compensation controls, fluorescence spread checks and isotype controls were utilized for single stains to ascertain compensation levels and assess nonspecific binding. The percentage of gated events expressing a specific marker was reported for each sample.

### 4.10. Enzyme-Linked Immunosorbent Assay

Plasma samples (Sham (mock, AMPKα1^F/F^); Sham (RIC, AMPKα1^F/F^); AMPKα1^F/F^ (TON); AMPKα1^F/F^ (TON+RIC); AMPKα1^F/F^ LysM^Cre^ (TON); AMPKα1^F/F^ LysM^Cre^ (TON+RIC), *n* = 5) were obtained upon sacrifice and diluted for the quantification of TNFα and IL-10 levels using ELISA following the manufacturer’s instructions (R&D Systems, Minneapolis, MN, USA; Biolegend, San Diego, CA, USA). Standards and samples were added to the wells and captured by the immobilized antibody. Following washing steps, an enzyme-linked polyclonal antibody specific for each cytokine was introduced, followed by the addition of a substrate solution resulting in the development of a colored product. The optical density of the color was measured at 450 nm. Sample concentrations were determined from the standard curve and adjusted for protein concentration.

### 4.11. Retinal Oxygen Measurement

In vivo oxygen measurements were conducted using a glass-electrode oxygen sensor (Unisense, Aarhus, Denmark). The sensor operates by detecting the diffusion of oxygen across a silicone membrane to an oxygen-reducing cathode. This cathode is polarized relative to an internal silver/silver chloride anode, and the resultant potential difference between the anode and cathode reflects the oxygen partial pressure. The sensor exhibits a linear response within the range of 0 to 1 Atm pO2, with minimal oxygen consumption (10 to 16 mol/s) and a response time of less than 1 s. Prior to implantation, the sensor underwent thorough prepolarization at −0.8 V for 24 h and calibration under conditions of maximum oxygen saturation (0.9% saline at 37 °C with air agitation), as well as zero oxygen (2% sodium ascorbate in 0.1 mol/L sodium hydroxide solution). Data acquisition was performed at a rate of 1 Hz using a portable computer, adhering to the Nyquist–Shannon sampling theorem. Anesthetized mice were secured in a stereotaxic frame (39463001; Leica, Buffalo Grove, IL, USA) equipped with a Cunningham mouse adaptor (39462950; Leica). Briefly, the head of the animal was stabilized, and a custom-made glass beveled needle (approximately 10 μm in diameter) was employed to create a narrow, self-sealing tunnel in the retina.

### 4.12. Transmission Electron Microscopy (TEM)

The TEM imaging and analysis of mouse optic nerve was performed by a TEM scientist at imaging cores—Electron Microscopy, the University of Arizona, Tucson, AZ, USA. For TEM analysis, the following perfusion protocols were implemented to ensure optimal ultrastructure detail of optic nerve. Mice were initially perfused with 100 mL of heparinized normal saline, followed by 200 mL of 2% paraformaldehyde and 2.5% glutaraldehyde in Millonig’s buffer. Subsequently, the optic nerves underwent osmication, dehydration and flat embedding in epoxy resin (Embed-812; Electron Microscopy Sciences, Hatfield, PA, USA) with cover slipping. Serial 40 nm sections were then cut and mounted onto Formvar-coated single-slotted grids. These grids were subsequently stained with 5% uranyl acetate in 50% methanol for 2 min and 0.5% lead citrate for 1 min before visualization using a JEM 1230 electron microscope (JEOL Ltd., Tokyo, Japan).

### 4.13. Statistical Analysis

Statistical analysis was performed using GrapdPad Prism (version 9.0.0) for OS X 193 (GraphPad Software, La Jolla, CA, USA). Each animal was considered an independent entity. The number of mice analyzed was indicated in the figure legend of each result. Multiple groups were compared using analysis of variance (ANOVA) test. The two-way ANOVA was used when there were two independent variables. Post hoc tests used were Dunnett’s and Tukey’s tests. Differences with *p* value of <0.05 were considered significant.

## 5. Conclusions

This study provides comprehensive evidence that RIC exerts neuroprotective effects in TON by preserving neuronal integrity, modulating microglial activation and regulating inflammatory responses through AMPKα1-dependent mechanisms. Additionally, RIC improves retinal oxygenation and promotes RGC survival and axonal regeneration, with some effects being independent of AMPK signaling. These findings suggest that RIC could be a promising therapeutic approach for mitigating the effects of TON, with AMPKα1 signaling playing a pivotal role in its efficacy. This study has the limitation that we did not include an AMPKα1^F/F^-LymsCre mock group as our study relied on a prior study for baseline data [10]. In schematic Figure 9, we showed that RIC therapy regulates immune cell signaling in eye trauma. Further study is warranted to find the molecular mechanisms.

## Figures and Tables

**Figure 1 ijms-25-13626-f001:**
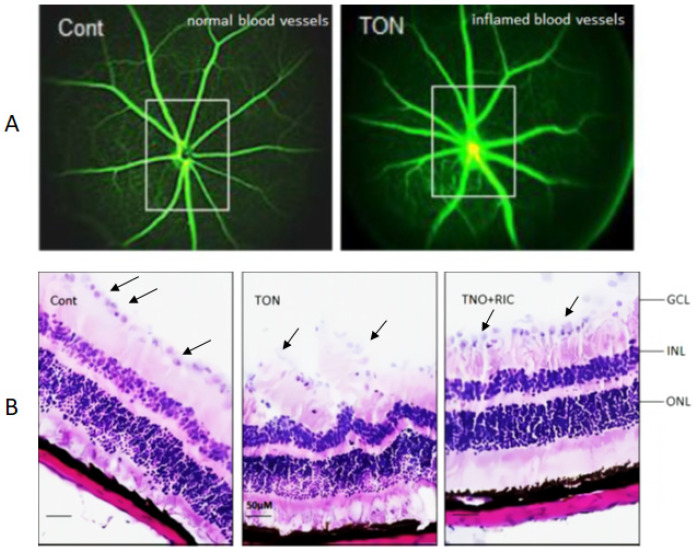
(**A**) Representative in vivo funduscopic fluorescein image from C56BL/6 mice showing inflammation in blood vessels in TON as compared with control eye. Intravenous fluorescein angiography of the mouse retina shows poor perfusion through attenuated vasculature (due to progression of the retinal degeneration) following TON. (**B**) H&E data showed increased neuronal cell death in ganglion cell layer in TON compared with control. However, the neuronal cell death is prevented with RIC treatment. Fluorescein angiography imaging (**A**) was captured within 5 mins of fluorescein dye injection through tail vein.

**Figure 2 ijms-25-13626-f002:**
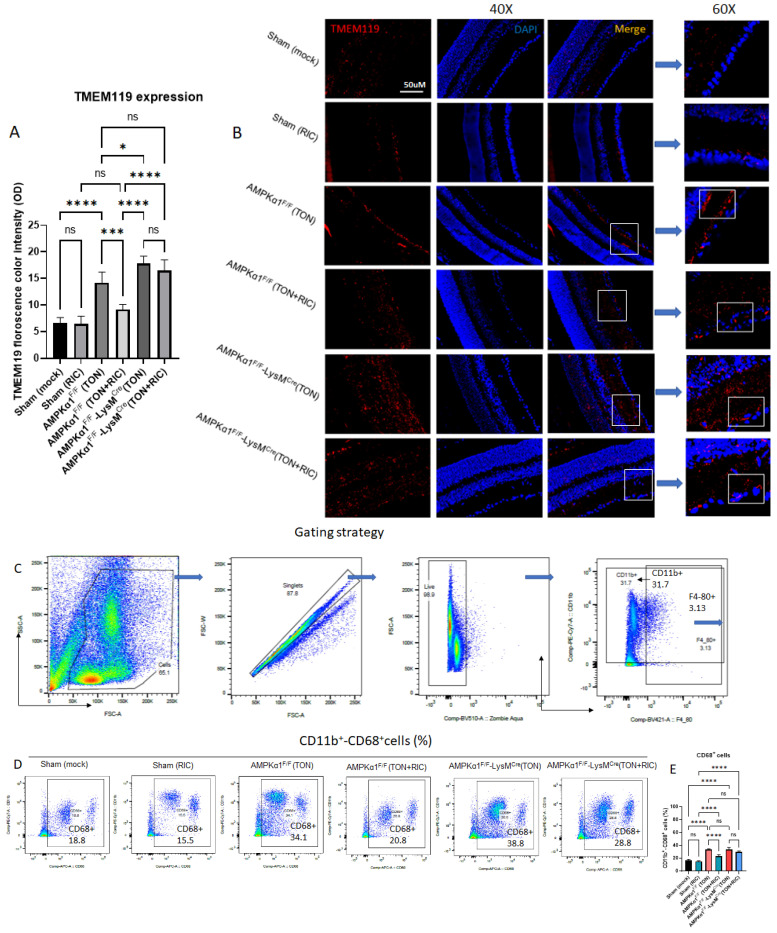
(**A**,**B**) Immunofluorescence staining showed microglial marker TMEM119 expression in mouse retina. TON (with AMPK) increases microglial activation, and RIC downregulated significantly. Myeloid pAMPKα1 KO group showed heightened microglial activation; notably, RIC demonstrated no significant effects. Florescence color intensity was measured by Image J software (NIH, https://imagej.net/ij/). White boxes show the TMEM119 expression in inner nuclear layer (INL) and GCL (ganglion cell layer) region of mouse eye. For Sham (mock) and Sham (RIC), both groups are regarded as AMPKα1^F/F^. (**C**–**I**) Representative pseudocolor and histograms of flow cytometry show the gating strategy for microglia/macrophages (CD11b+_F4/80+) and CD68+ and CD206+ expressing microglia in blood. Bar graph summarizing the cell counts of microglia (M1/M2) in the blood after 5 days of TON. Red, TMEM119 (activated microglial marker); Blue, DAPI. We used 6 experimental groups, Sham (mock); Sham (RIC); AMPKα1^F/F^ (TON); AMPKα1^F/F^ (TON+RIC); AMPKα1^F/F^ LysMCre (TON); AMPKα1^F/F^ LysMCre (TON+RIC). Differences among experimental groups were determined by analysis of variance (one-way ANOVA) followed by Newman–Keuls multiple comparison tests. The results represent the means ± SEM of fold changes (*n* = 5). * *p* < 0.05, *** *p* < 0.001, **** *p* < 0.0001. ns, non-significant. For Sham (mock) and Sham (RIC), both groups are regarded as AMPKα1^F/F^.

**Figure 3 ijms-25-13626-f003:**
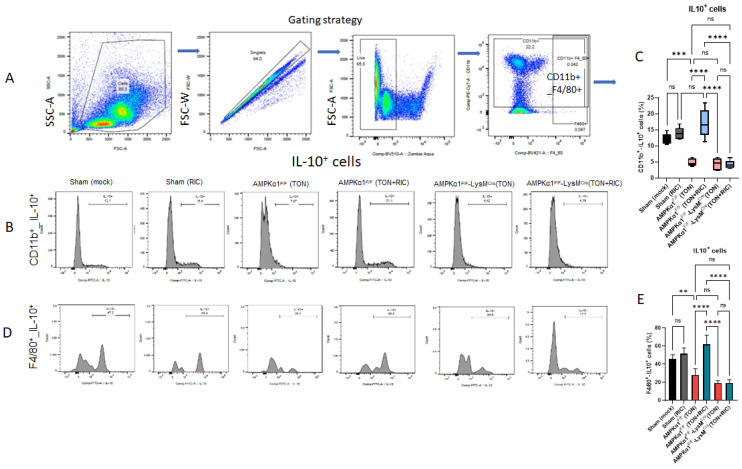
Effect of RIC on IL10 and neutrophil expression following TON. (**A**,**B**,**D**,**F**) Representative pseudocolor and histograms of flow cytometry show the gating strategy for microglia/macrophages (CD11b+_IL10+, F4/80+_IL10+ and CD68+_IL10+) and CD68+_LY6G+-expressing neutrophils in blood. (**C**,**E**,**G**,**H**) Representative bar graph summarizing the cell counts of IL10+ and Ly6G+ in the blood after 5 days of TON. Six experimental groups included Sham (mock); Sham (RIC); AMPKα1^F/F^ (TON); AMPKα1^F/F^ (TON+RIC); AMPKα1^F/F^ LysMCre (TON); AMPKα1^F/F^ LysMCre (TON+RIC). Differences among experimental groups were determined by analysis of variance (one-way ANOVA) followed by Newman–Keuls multiple comparison tests. The results represent the means ± SEM of fold changes (*n* = 5). ** *p* < 0.01.*** *p* < 0.001, **** *p* < 0.0001. ns, non-significant. For Sham (mock) and Sham (RIC), both groups are regarded as AMPKα1^F/F^.

**Figure 4 ijms-25-13626-f004:**
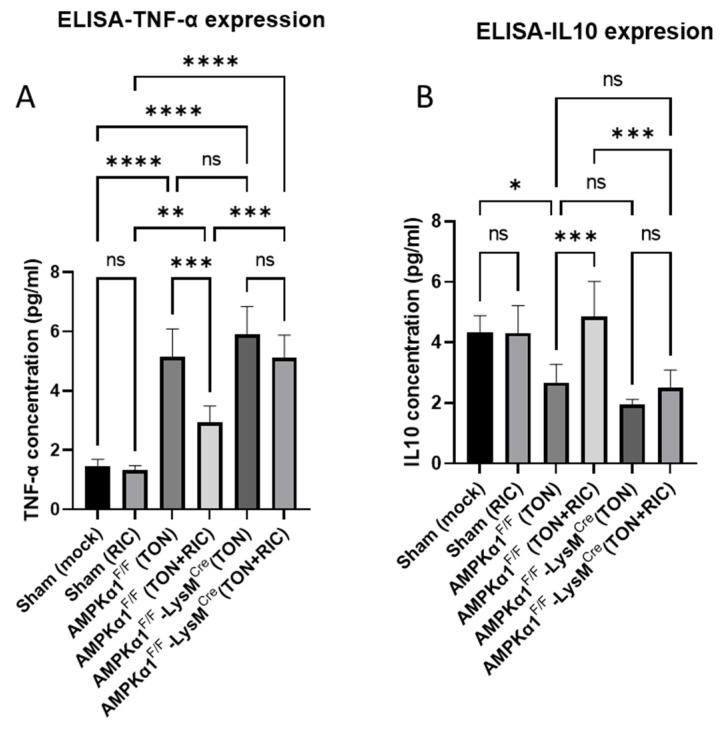
The effect of RIC on TON induced pro-inflammatory signaling. (**A**,**B**) ELISA results in blood plasma showing TNF and IL10 expression. Fluorescence color intensity was measured by Image J software. We used 6 experimental group, Sham (mock); Sham (RIC); AMPKα1^F/F^ (TON); AMPKα1^F/F^ (TON+RIC); AMPKα1^F/F^ LysM^Cre^ (TON); AMPKα1^F/F^ LysM^Cre^ (TON+RIC). Differences among experimental groups were determined by analysis of variance (one-way ANOVA) followed by Newman–Keuls multiple comparison tests. The results represent the means ± SEM of fold changes (*n* = 5). * *p* < 0.05, ** *p* < 0.01, *** *p* < 0.001, **** *p* < 0.0001. ns, non-significant. For Sham (mock) and Sham (RIC), both groups are regarded as AMPKα1^F/F^.

**Figure 5 ijms-25-13626-f005:**
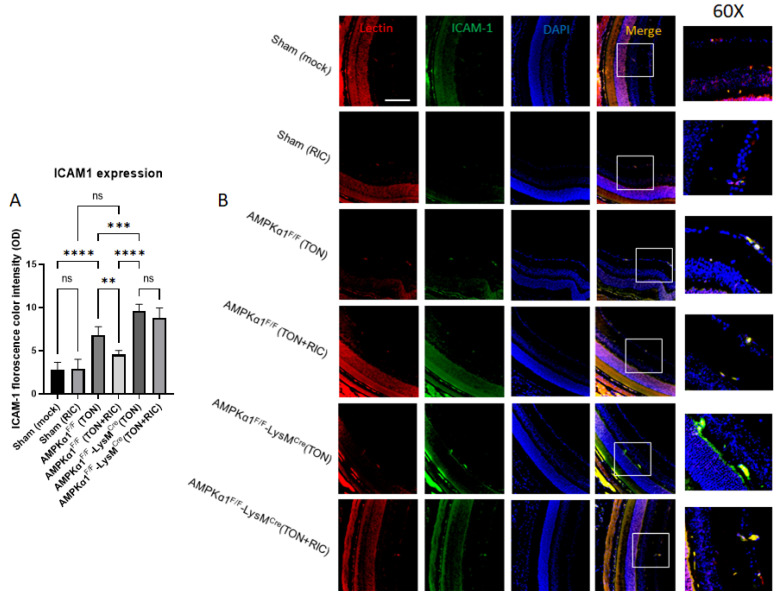
Effect of RIC on TON induced pro-inflammatory signaling. (**A**,**B**) The effect of RIC on ICAM-1 expression assessed by immunofluorescence and (**C**,**D**) ICAM1 Protein expression was checked by Western blot. Fluorescence color intensity as well as western blot band intensity was measured by Image J software (NIH, https://imagej.net/ij/). We used 6 experimental groups, Sham (mock); Sham (RIC); AMPKα1^F/F^ (TON); AMPKα1^F/F^ (TON+RIC); AMPKα1^F/F^ LysM^Cre^ (TON); AMPKα1^F/F^ LysM^Cre^ (TON+RIC). Differences among experimental groups were determined by analysis of variance (one-way ANOVA) followed by Newman–Keuls multiple comparison tests. The results represent the means ± SEM of fold changes (*n* = 5). * *p* < 0.05, ** *p* < 0.01, *** *p* < 0.001, **** *p* < 0.0001. ns, non-significant. Scale bar 50 μm. For Sham (mock) and Sham (RIC), both groups are regarded as AMPKα1^F/F^.

**Figure 6 ijms-25-13626-f006:**
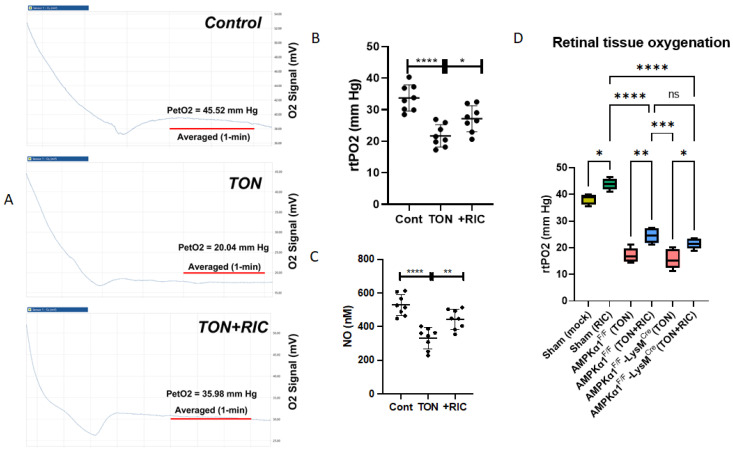
(**A**–**D**) Effect of RIC therapy on retinal oxygenation in TON. Oxygen levels were analyzed with UniSense sensor system (Sweden). We used 6 experimental groups, Sham (mock); Sham (RIC); AMPKα1F/F (TON); AMPKα1F/F (TON+RIC); AMPKα1F/F LysM^Cre^ (TON); AMPKα1F/F LysM^Cre^ (TON+RIC). Differences among experimental groups were determined by analysis of variance (one-way ANOVA) followed by Newman–Keuls multiple comparison tests. The results represent the means ± SEM of fold changes (*n* = 5). * *p* < 0.05, ** *p* < 0.01, *** *p* < 0.001, **** *p* < 0.0001. ns, non-significant. For Sham (mock) and Sham (RIC), both groups are regarded as AMPKα1^F/F^.

**Figure 7 ijms-25-13626-f007:**
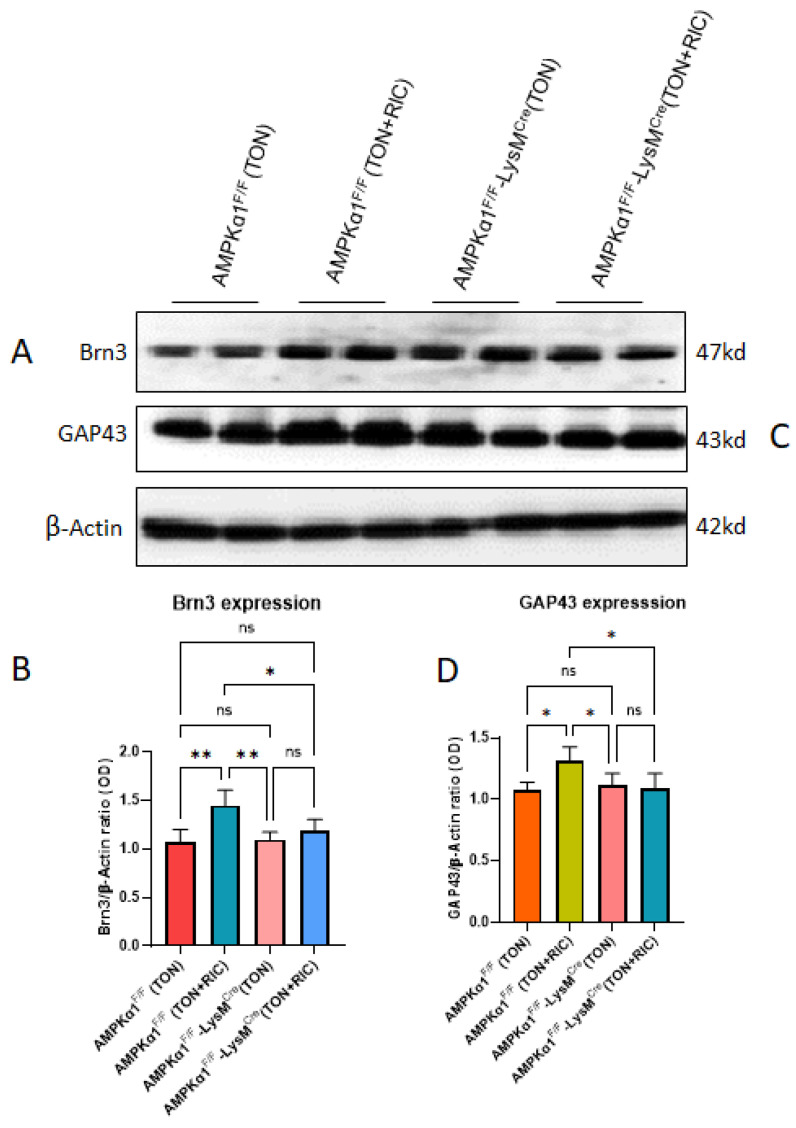
(**A**–**D**) Effect of RIC therapy on TON retina. Western blot analysis demonstrated significant changes in protein expression level of Brn3a and GAP43 between TON+RIC and TON group. Densitometry analysis was carried out by Image J software (NIH, https://imagej.net/ij/). We used 4 experimental groups, AMPKα1F/F (TON); AMPKα1F/F (TON+RIC); AMPKα1F/F LysM^Cre^ (TON); AMPKα1F/F LysM^Cre^ (TON+RIC). Differences among experimental groups were determined by analysis of variance (one-way ANOVA) followed by Newman–Keuls multiple comparison tests. The results represent the means ± SEM of fold changes (*n* = 5). * *p* < 0.05, ** *p* < 0.01. ns, non-significant.

**Figure 8 ijms-25-13626-f008:**
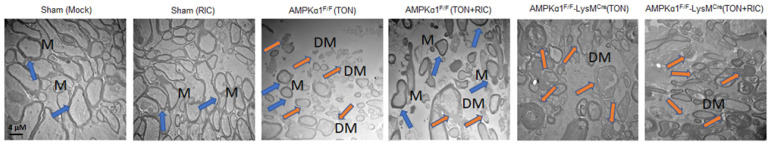
Representative ultrastructural features of axonal injury in traumatic optic neuropathy. Electron micrographs are taken across the longitudinal plane through the injury front and show a range of axoplasmic, axolemmal and myelin sheath abnormalities. RIC therapy attenuated this degenerating process in TON. We used 6 experimental groups, Sham (mock); Sham (RIC); AMPKα1F/F (TON); AMPKα1F/F (TON+RIC); AMPKα1F/F LysM^Cre^ (TON); AMPKα1F/F LysM^Cre^ (TON+RIC). Scale bar 4 μm. For Sham (mock) and Sham (RIC), both groups are regarded as AMPKα1^F/F^.

**Figure 9 ijms-25-13626-f009:**
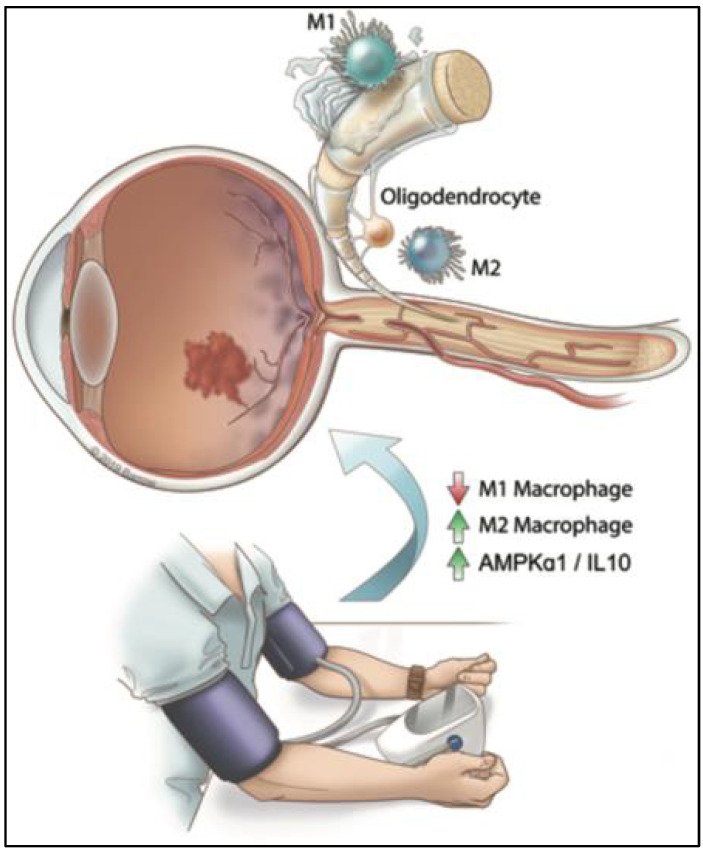
Schematic representation demonstrating increased M1-type macrophages causing inflammation and demyelination of optic nerve (ON) in TON. Our hypothesis demonstrates that RIC therapy activates AMPKα1 to modulate macrophage polarization toward M2-type anti-inflammatory macrophages that protect demyelination of downregulated ON.

## Data Availability

Data are contained in the article.

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
