# Peer review of "Remote Ischemic Post-Conditioning (RIC) Mediates Anti-Inflammatory Signaling via Myeloid AMPKα1 in Murine Traumatic Optic Neuropathy (TON)"

_ijms, 2024, doi:10.3390/ijms252413626_

Round 1

Reviewer 1 Report

Comments and Suggestions for Authors

The paper investigates the neuroprotective effects of RIC on TON and the critical role of myeloid-specific AMPKα1 signaling in modulating inflammation and neuronal survival. Using a murine model, the study reveals that RIC therapy mitigates inflammation and neuronal degeneration primarily through activation of the AMPKα1 pathway in myeloid cells. Meanwhile, they also showed that RIC induced retinal oxygenation in an AMPK-independent manner. These findings underscore AMPKα1’s necessity in harnessing RIC's benefits, while highlighting the importance of other signaling pathways. This study is answering an important question in the field, but the experiment design and results presentation need to be polished to reach the publication standard to IJMS.

Comments:

1. There is no scale bar in any fluorescent images (Figures 2 and 5).

2. In Figure 2A, what is the meaning of the white boxes in the last column? Also, there should be a box in the second last column showing where the enlarged images are zoomed from. Figure 5A shows a good example of that.

3. Why do the authors separate the figures into several sections? For example, Figure 2 A-B, C-E, F-I are all separate. This is highly confusing for the readers, and must be revised.

4. Please explain why the authors chose the current 6 groups in their experiment.

(1) Is the sham group WT mice or AMPKa1f/f mice? If WT mice, what is the point of including AMPKa1f/f mice in the experiment?

(2) Why is there no TON treatment group with sham mice?

(3) Why there is no mock group (no TON treatment) with AMPKa1f/f or AMPKa1f/f-LymsCre mice?

To have a fair comparison, at least one additional group of AMPKa1f/f-LymsCre mice without TON or RIC needs to be provided, otherwise, we cannot estimate the baseline impact of AMPKa1 KO on the mouse retina.

5. The authors claim that “TON did not significantly affect myeloid expression of either CD11b+-CD206+ or F4/80+- CD206+ in either WT or myeloid-specific AMPKα1 knockout mice” (Line 183-185). Without a WT TON group or AMPKa1 KO mock group, how do we even compare the TON impact on these conditions?

6. The authors provided a lot of flow cytometry data without clearly explaining the meaning of it.

(1) Is CD11b⁺CD68⁺ an appropriate marker for pro-inflammatory macrophages? Would CD11b⁺CD86⁺ be more suitable for this purpose?

(2) Why do the authors measure CD11b+ and F4/80+ cells separately? Please explain the difference in the text. Why F4/80+CD68+ cell data is absent while F4/80+CD206+, F4/80+IL10+ data are presented?

(3) What is the difference between CD206+ cells and IL-10+ cells if they are both anti-inflammatory?

7. The discussion section spans five pages without subtitles, which can make it challenging to follow. Consider adding subtitles to structure the discussion and improve readability.

Reviewer 2 Report

Comments and Suggestions for Authors

Akhter et al. found that remote ischemic post-conditioning mediates neuroprotection via myeloid AMPKα1 signaling in murine traumatic optic neuropathy. It is interesting and well-written with having important information. However, the data presentation and some data's poor images make the manuscript have less impact. This should be addressed. Below are the points.

Figure 1A's data is permanent effects? or transient effects? This should be addressed with experiments based on time points.

Figure 1's HnE images should be placed in the same axis (photoreceptors to ganglion cells vertically) and include quantification data.

Figure 2A and Figure 5A's images should be placed in the same axis (photoreceptors to ganglion cells vertically).

Figure 5C, 7A, and 7C's western blot images' qualities are poor. This should be replaced.

Colors of the bar in each figure are random. It is unfriendly for the reader to follow up the entire study. This should be matched. 

In Figure 8, EM's scale bar is missing?

Round 2

Reviewer 1 Report

Comments and Suggestions for Authors

The revised manuscript made great progress and addressed most of my previous comments. However, I do have a few follow-up comments.

1. I understand the authors' rationale for separating Figure 2 into multiple sections to illustrate different aspects of the hypothesis. However, I suggest consulting the editorial office to confirm that this approach aligns with the journal's publication guidelines to avoid potential layout issues during production.

2. To enhance clarity and avoid potential misinterpretation, I recommend renaming the groups as follows:

  • Change “Sham (mock)” to “AMPKa1F/F (mock)”
  • Change “Sham (RIC)” to “AMPKa1F/F (RIC)”

This naming convention will more accurately reflect that the "sham" groups also carry the AMPKa1F/F genotype, minimizing confusion regarding group designations.

3. I acknowledge the challenges involved in including an additional AMPKa1F/F-LymsCre mock group due to breeding limitations, as well as the reliance on prior studies (e.g., Vaibhav et al., 2018) for baseline data. However, given the variability between experimental setups, providing separate baseline data remains essential to comprehensively interpret the study’s findings. I still consider this a major limitation, and I recommend adding a discussion on this issue in the Discussion section. Highlighting this limitation transparently will strengthen the manuscript's scientific rigor.

Author Response

  1. I understand the authors' rationale for separating Figure 2 into multiple sections to illustrate different aspects of the hypothesis. However, I suggest consulting the editorial office to confirm that this approach aligns with the journal's publication guidelines to avoid potential layout issues during production.

Thank you for the comment.

  1. To enhance clarity and avoid potential misinterpretation, I recommend renaming the groups as follows:
  • Change “Sham (mock)” to “AMPKa1F/F (mock)”
  • Change “Sham (RIC)” to “AMPKa1F/F (RIC)”

This naming convention will more accurately reflect that the "sham" groups also carry the AMPKa1F/F genotype, minimizing confusion regarding group designations.

Thank you and we have incorporated the changes in Figure legend as well as in Methods section. Changes are highlighted in yellow

  1. I acknowledge the challenges involved in including an additional AMPKa1F/F-LymsCre mock group due to breeding limitations, as well as the reliance on prior studies (e.g., Vaibhav et al., 2018) for baseline data. However, given the variability between experimental setups, providing separate baseline data remains essential to comprehensively interpret the study’s findings. I still consider this a major limitation, and I recommend adding a discussion on this issue in the Discussion section. Highlighting this limitation transparently will strengthen the manuscript's scientific rigor.

We agree with reviewer's advice and we have mentioned this in discussion section.

Reviewer 2 Report

Comments and Suggestions for Authors

The revised manuscript addresses the raised comments.

All retinal section images should have the same axis. It is randomly presented. Vertically, from ONL to GCL, it should be presented.

Sagittal section-based images show a limited window. Flat-mount could be used to examine inflammatory cell changes in this model.

The others are addressed based on the previous comments.

Author Response

All retinal section images should have the same axis. It is randomly presented. Vertically, from ONL to GCL, it should be presented.

We are sorry for the inconvenience. The labtech took the immunofluorescence images under the microscope. Immunofluorescence slides can not be stored more than 3 days as color starts fading. Now we can not plan to do same experiment again to get right pictures as It will take weeks to complete the work. We are mainly showing the GCL layer and we have mentioned clearly. We will follow the reviewer's advice in our ongoing study.

Sagittal section-based images show a limited window. Flat-mount could be used to examine inflammatory cell changes in this model.

We agree with the reviewer's suggestions and agree that we should examine flat-mount sections. Unfortunately, we do not have enough mice to do the experiment. we are planning to do the same in our extended study.

The others are addressed based on the previous comments.

Thank you.